# Disparate bone anabolic cues activate bone formation by regulating the rapid lysosomal degradation of sclerostin protein

Nicole R Gould[1]*, Katrina M Williams[1], Humberto C Joca[2], Olivia M Torre[1], James S Lyons[1], Jenna M Leser[1], Manasa P Srikanth[3], Marcus Hughes[1], Ramzi J Khairallah[4], Ricardo A Feldman[3], Christopher W Ward[1]*, Joseph P Stains[1]*

[1]Department of Orthopaedics, University of Maryland School of Medicine, Baltimore, United States; [2]Center for Biomedical Engineering and Technology, University of Maryland School of Medicine, Baltimore, United States; [3]Department of Microbiology and Immunology, University of Maryland School of Medicine, Baltimore, United States; [4]Myologica, LLC, New Market, United States

**Abstract** The downregulation of sclerostin in osteocytes mediates bone formation in response to mechanical cues and parathyroid hormone (PTH). To date, the regulation of sclerostin has been attributed exclusively to the transcriptional downregulation of the *Sost* gene hours after stimulation. Using mouse models and rodent cell lines, we describe the rapid, minute-scale post-translational degradation of sclerostin protein by the lysosome following mechanical load and PTH. We present a model, integrating both new and established mechanically and hormonally activated effectors into the regulated degradation of sclerostin by lysosomes. Using a mouse forelimb mechanical loading model, we find transient inhibition of lysosomal degradation or the upstream mechano-signaling pathway controlling sclerostin abundance impairs subsequent load-induced bone formation by preventing sclerostin degradation. We also link dysfunctional lysosomes to aberrant sclerostin regulation using human Gaucher disease iPSCs. These results reveal how bone anabolic cues post-translationally regulate sclerostin abundance in osteocytes to regulate bone formation.

**\*For correspondence:**
ngould@som.umaryland.edu (NRG);
ward@som.umaryland.edu (CWW);
jstains@som.umaryland.edu (JPS)

## Introduction

The osteocyte-derived protein, sclerostin, is a fundamentally important inhibitor of bone formation, with decreases in abundance mediating mechanically and hormonally induced bone formation. Sclerostin (gene name *Sost*) is a secreted 27 kDa glycoprotein that inhibits the differentiation and activity of bone-forming osteoblasts by antagonizing the Wnt/β-catenin signaling pathway (*Baron and Gori, 2018*; *Drake and Khosla, 2017*). Genetic deletion of the *Sost* gene in mice results in extraordinarily high bone mass (*Li et al., 2008*). In humans, mutations in the *SOST* gene underlie high bone mass and bone overgrowth in patients with sclerosteosis and van Buchem disease (*Balemans et al., 2002*; *Balemans et al., 2001*; *Appelman-Dijkstra et al., 1993*). Accordingly, regulating sclerostin bioavailability has tremendous therapeutic potential for conditions of low bone mass, such as osteoporosis. Indeed, targeting sclerostin protein with neutralizing antibodies is incredibly effective at increasing bone mass, and Romosozumab, a humanized monoclonal antibody targeting sclerostin, has been FDA approved to treat osteoporosis in post-menopausal women at a high risk for fracture (*McClung, 2017*; *Bandeira et al., 2017*). Despite this critical role for sclerostin in skeletal

homeostasis and its therapeutic potential, there are substantial gaps with respect to the molecular control of this key regulatory protein.

In response to bone mechanical loading, osteocytes sense and respond to fluid shear stress (FSS) in the lacunar canicular network by ultimately decreasing sclerostin protein abundance, de-repressing Wnt/β-catenin signaling, and unleashing osteoblast differentiation and bone formation. When administered intermittently, parathyroid hormone (PTH) causes net bone formation in part by decreasing sclerostin (*Keller and Kneissel, 2005*; *Bellido et al., 2005*). This has been exploited in the clinic through the established osteoanabolic drug, teriparatide (PTH, amino acids 1–34). Despite their clinical application, little is known about how mechanical load and PTH, two disparate bone anabolic signals, directly regulate sclerostin protein. To date, the regulation of sclerostin protein abundance has been attributed to the transcriptional downregulation of the *Sost* gene that occurs on an hour timescale after mechanical load or PTH exposure (*Sebastian and Loots, 2017*; *Wein, 2018*; *Bonnet et al., 2012*; *Bonnet et al., 2009*; *Meakin et al., 2014*).

Using a recently established osteocyte-like cell line, Ocy454 cells, which is one of the few cell lines that reliably express detectable sclerostin protein (*Wein et al., 2015*; *Spatz et al., 2015*), we previously described a mechano-transduction pathway that regulates osteocyte sclerostin protein abundance in response to FSS in vitro (*Figure 1A*; *Lyons et al., 2017*; *Williams et al., 2020*). Using this in vitro model, we found that osteocyte mechano-signaling required a subset of detyrosinated microtubules, which transduce load signals to activate NADPH oxidase 2 (NOX2), which produces reactive oxygen species (ROS) signals. These ROS signals then elicit a transient receptor potential vanilloid 4 (TRPV4)-dependent primary calcium ($Ca^{2+}$) influx. Calcium/calmodulin-dependent kinase II (CaMKII) is activated in response to this primary $Ca^{2+}$ influx and is required for reduction of osteocyte sclerostin protein abundance (*Figure 1A*). While these discoveries integrated with and extended several established models of the osteocyte mechanical response (*Thompson et al., 2012*; *Schaffler et al., 2014*; *Geoghegan et al., 2019*; *Baik et al., 2013*), we found the loss of sclerostin protein was surprisingly rapid (minute scale) and was likely wholly distinct from the well-characterized transcriptional regulation of the *Sost* gene, which occurs on the hour timescale (*Sebastian and Loots, 2017*; *Wein, 2018*; *Bonnet et al., 2012*; *Bonnet et al., 2009*; *Meakin et al., 2014*). Despite its physiologic significance, little is known about the post-translational control of sclerostin protein. Additionally, given the in vitro nature of our prior work on this pathway, the contribution of this mechano-transduction pathway to in vivo bone mechano-responsiveness remained unresolved. Here, we examined the kinetics of sclerostin protein degradation in vitro and in vivo, the mechanism regulating the rapid decline in osteocyte sclerostin protein, its relevance in vivo in bone physiology during ulnar load, and its dysfunction in skeletal disease. We also extended the observation of the FSS-induced rapid degradation of sclerostin protein to PTH, another clinically relevant bone anabolic signal.

## Results

### Sclerostin protein is rapidly lost after mechanical load in vitro and in vivo

Our prior work showed sclerostin protein disappearance on a minute timescale following FSS in Ocy454 cells, an effect that required activation of CaMKII (*Figure 1A*; *Lyons et al., 2017*; *Williams et al., 2020*). To more precisely characterize the dynamics of mechanically stimulated sclerostin protein loss in osteocytes, we examined two sclerostin expressing cell lines: Ocy454 osteocytes and UMR106 osteosarcoma cells. Consistent with our previous work, we observed the mechano-activated increase in CaMKII phosphorylation and decrease in sclerostin protein abundance in both Ocy454 osteocytes (*Figure 1B*) and UMR106 osteosarcoma cells (*Figure 1C*) within 5 min of the acute application of FSS. To establish the relevance of a load-induced loss of sclerostin protein in vivo, we examined if this rapid loss of sclerostin abundance, as determined by western blot or sclerostin immunofluorescence in sectioned bone, occurred following an acute bout of ulnar loading. Sclerostin protein abundance was assessed in 16 week old female or 17 week old male mice subjected to a single bout of ulnar load. Ulnae were harvested 5 min post-load, a time frame consistent with in vitro observations, and osteocyte-enriched cortical bone lysates were profiled by western blotting or ulnae were sectioned to evaluate sclerostin-positive osteocytes by

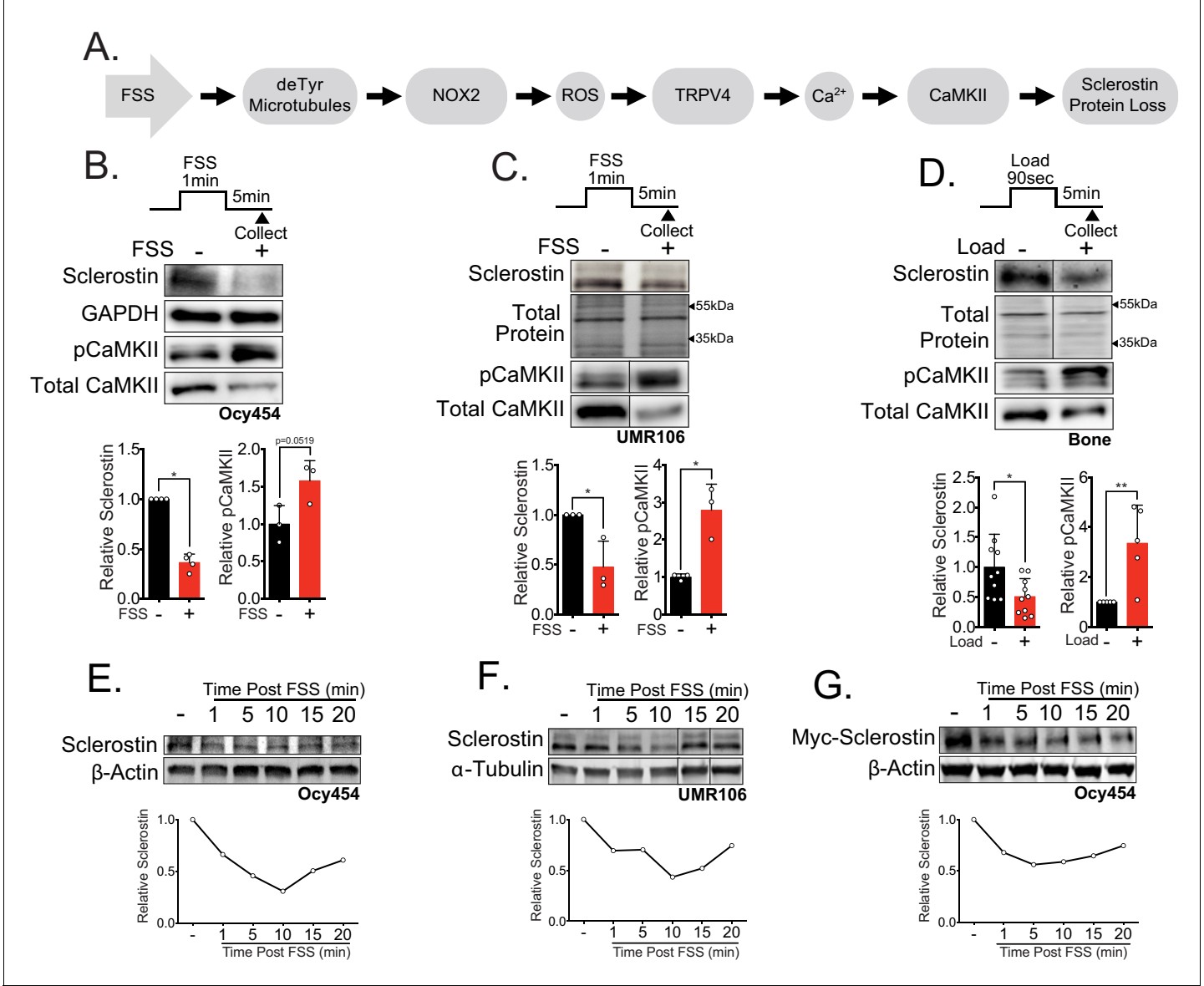

**Figure 1.** Sclerostin protein is rapidly degraded after mechanical stimulus in vitro and in vivo. (A) FSS causes the rapid loss of sclerostin protein through a number of molecular mediators. (B) Ocy454 cells (n = 3–4) or (C) UMR106 cells (n = 3) were exposed to 1 min of FSS at 4 dynes/cm$^2$ and lysed 5 min post-flow. Western blots were probed for sclerostin, GAPDH, pCaMKII, and total CaMKII. (D) Sixteen week old female C57Bl/6 mice were ulnar loaded (1800 με, 90 s, 2 Hz), cortical osteocyte-enriched lysates isolated 5 min post-load, and western blots probed for sclerostin (n = 10 mice), pCaMKII, and total CaMKII (n = 5 mice). Sclerostin abundance relative to the loading control or pCaMKII relative to total CaMKII was quantified. (E) Ocy454 cells with endogenous sclerostin (n = 2), (F) UMR106 cells with endogenous sclerostin (n = 4), or (G) Ocy454 cells transfected with Myc-tagged sclerostin (n = 1) were subjected to 5 min of FSS at 4 dynes/cm$^2$ and lysed at the indicated times post-flow. Western blots were probed for sclerostin and β-actin. A representative time course is shown for each. Sclerostin abundance relative to the loading control was quantified. For each antibody, western blots are from a single gel and exposure; a vertical black line indicates removal of irrelevant lanes. Graphs depict mean ± SD. *p<0.05, **p<0.01 by unpaired two-tailed t-tests (B–D).

The online version of this article includes the following figure supplement(s) for figure 1:

**Figure supplement 1.** Rapid loss of sclerostin protein occurs in osteocytes in vivo and in vitro.

immunofluorescence. As occurred in vitro, sclerostin protein abundance was rapidly reduced and CaMKII activated in loaded versus contralateral non-loaded limbs (*Figure 1D*), and the percent of sclerostin positive osteocytes was decreased in loaded limbs versus contralateral non-loaded limbs

(*Figure 1—figure supplement 1A*), establishing that rapid sclerostin protein loss also occurs in vivo following a single bout of mechanical stimulus.

A temporal assessment of sclerostin protein abundance following FSS in Ocy454 cells (*Figure 1E*) and UMR106 cells (*Figure 1F*) revealed a decrease in sclerostin protein as early as 1 min post-FSS, with the nadir of protein expression occurring at 10 min, followed by a subsequent rebound. This rapid regulation strongly suggested post-translational control of sclerostin abundance. Consistent with the possibility of post-translational control of sclerostin, we recently reported that a single 5 min bout of FSS, as used here, does not decrease *Sost* mRNA levels despite a decrease in sclerostin protein (*Williams et al., 2020*). To confirm the post-translational control of sclerostin, Ocy454 cells were transfected with either a myc- or GFP-tagged sclerostin expression vector under control of a CMV promoter, which is not controlled by the same transcriptional elements that regulate *Sost* transcription. Like endogenous sclerostin, overexpressed sclerostin protein was rapidly and transiently decreased by FSS (*Figure 1G*, *Figure 1—figure supplement 1B*). The regulated degradation did not extend to all proteins as pro-collagen type 1α1 abundance was unchanged following FSS in Ocy454 cells (*Figure 1—figure supplement 1C*), supporting some level of specificity for the regulated degradation of sclerostin.

## PTH1-34 treatment also causes rapid sclerostin protein loss

Like mechanical loading, intermittent administration of PTH stimulates bone remodeling and results in net bone formation (*Compston, 2007*). Both chronic and intermittent PTH treatment decrease osteocyte *Sost* gene expression and protein abundance hours after exposure *Sebastian and Loots, 2017*; however, rapid loss of sclerostin protein has not been reported. In Ocy454 cells, sclerostin protein abundance decreased after 30 min of PTH treatment (*Figure 1—figure supplement 1D*), with no change in pro-collagen type 1α1 (*Figure 1—figure supplement 1E*). Examining the post-translational regulation of sclerostin protein abundance in GFP-sclerostin transfected Ocy454 cells, we show PTH treatment caused sclerostin loss over a 30 min time period (*Figure 2A*), with no change in *Sost* mRNA levels after this 30 min treatment (*Figure 2—figure supplement 1*). To confirm that rapid regulation occurs in an intact bone with endogenous sclerostin, dissected tibiae were

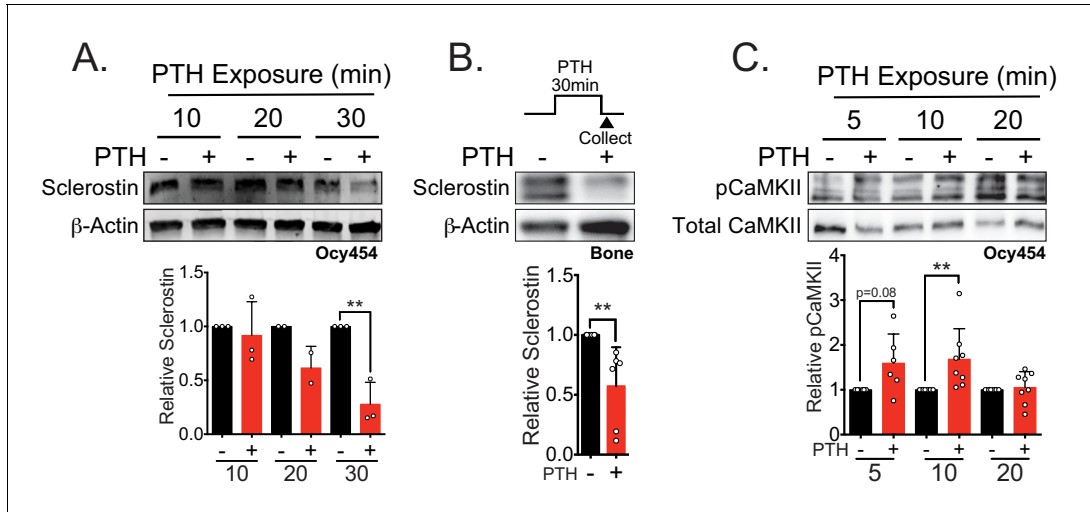

**Figure 2.** Sclerostin protein is rapidly degraded after PTH exposure in vitro and ex vivo. (**A**) Ocy454 cells transfected with GFP-sclerostin were treated with vehicle or PTH (1–34) (10 nM) for the indicated time and were lysed. Western blots were probed for sclerostin and β-actin (n = 2–3). (**B**) Dissected tibiae flushed of marrow were treated with vehicle or PTH (1–34) (10 nM) for 30 min ex vivo and homogenized. Western blots were probed for sclerostin and β-actin (n = 6 mice). (**C**) Ocy454 cells were treated with vehicle or PTH (1–34) (10 nM) for the indicated time and were lysed. Western blots were probed for pCaMKII and total CaMKII (n = 6–8). Graphs depict mean ± SD. *p<0.05, **p<0.01 by two-way ANOVA with Holm–Sidak post hoc correction (A, C) or unpaired two-tailed t-test (B).

The online version of this article includes the following figure supplement(s) for figure 2:

**Figure supplement 1.** Acute treatment with PTH does not affect *Sost* mRNA hours later.

treated with PTH ex vivo, which decreased sclerostin protein abundance in the same minute-scale timeframe (*Figure 2B*).

Given that both FSS and PTH treatment rapidly and post-translationally regulate sclerostin abundance, we next wanted to examine common signal transducers between the two stimuli. Like FSS, PTH may also regulate CaMKII activation in bone cells (*Quinn et al., 2000*; *Williams et al., 2016*). As with FSS (*Lyons et al., 2017*; *Williams et al., 2020*), PTH exposure increased phospho-CaMKII abundance (*Figure 2C*), representing a possible intersection point in the signaling cascades used by these two anabolic stimuli.

In total, while the kinetics sclerostin downregulation following PTH treatment were slower than FSS, the loss of sclerostin protein was still relatively rapid and occurred with overexpressed protein, supporting that both mechanical load and PTH regulates post-translational control of sclerostin, possibly through CaMKII activation.

## Sclerostin is degraded through the lysosome after exposure to bone anabolic stimuli

Next, we examined the mechanism by which sclerostin protein was so rapidly reduced. The post-translational loss of sclerostin protein shortly after exposure to a bone anabolic stimulus likely occurs through either rapid protein degradation or protein secretion. Accordingly, we interrogated the route of rapid sclerostin loss after anabolic stimuli by blocking protein degradation or protein secretion pathways using different pharmacological inhibitors. In these degradation assays, cells were treated with cycloheximide (CHX) to prevent de novo protein synthesis, as well as either the lysosome inhibitor, bafilomycin A1, the proteasome inhibitor, MG-132, or the secretion inhibitor, brefeldin A. Subsequently, the cells were exposed to 5 min of FSS or 30 min of PTH, and sclerostin protein abundance monitored by western blot. In Ocy454 cells transfected with GFP-sclerostin, inhibition of lysosomal function with bafilomycin A1 prevented both FSS- and PTH-induced degradation of sclerostin protein (*Figure 3A,B*), whereas inhibition of proteasomal degradation with MG-132 or secretion with brefeldin A had no effect on the FSS-induced decrease in sclerostin (*Figure 3A*). Likewise, inhibition of lysosomal activity with bafilomycin A1 or leupeptin prevented FSS-induced sclerostin degradation in UMR106 osteosarcoma cells (*Figure 3—figure supplement 1A*). Notably, in the absence of a stimulus, the half-life of sclerostin protein is about 3 hr (*Figure 3—figure supplement 1B*). This is in stark contrast to the minute-scale degradation observed after stimulation with FSS or PTH, confirming that the kinetics of sclerostin degradation are altered in response to bone anabolic cues.

In support of this notion of lysosomal degradation of sclerostin, many secreted N-linked mannose-6-phosphate and GlcNAC-modified glycoproteins, like sclerostin (*Brunkow et al., 2001*), are targeted to the lysosome to regulate abundance (*Braulke and Bonifacino, 2009*). Typically, these proteins contain conserved lysosomal signal sequences, particularly the Asn-X-Ser/Thr sites that are subjected to N-linked glycosylation (*Braulke and Bonifacino, 2009*). In silico analysis of the sclerostin protein amino acid sequence identified at least two putative sites for N-linked glycosylation (Asn-X-Ser/Thr) and three Tyr-X-X-Φ lysosome targeting motifs that are conserved across species (*Figure 3C* and *Figure 3—figure supplement 1C*).

## Sclerostin protein co-localizes with lysosomal markers

To validate that sclerostin can be targeted to the lysosome, we examined the sub-cellular distribution of sclerostin and lysosomes in cultured Ocy454 cells. Both endogenous and exogenous sclerostin were found in distinct puncta in Ocy454 cells (*Figure 4A*). In live Ocy454 cells, GFP-sclerostin was co-localized with acidic vesicles identified by LysoTracker and co-localized with lysosomes labeled by siR-lysosome (*Figure 4B*). In 3D projections of Z-stacks, it is clear that sclerostin co-localizes with lysosomes identified by siR-lysosome, but not all sclerostin is contained in lysosomes and not all lysosomes contain sclerostin (*Figure 4—figure supplement 1*). In fixed cells, endogenous sclerostin co-localized with p62/sequestosome-1 protein, an autophagy cargo adapter protein that shuttles proteins for lysosomal degradation (*Figure 4C*). That sclerostin is found localized with lysosomal adapter proteins and is degraded by the lysosome is consistent with the recent finding that sclerostin is found in extracellular vesicles positive for the lysosome-associated protein LAMP1 (*Morrell et al., 2018*). Interestingly, sclerostin was also co-detected with Rab27a in Ocy454 cells (*Figure 4C*). Rab27a is a small GTPase that is a master regulator of the trafficking, docking, and

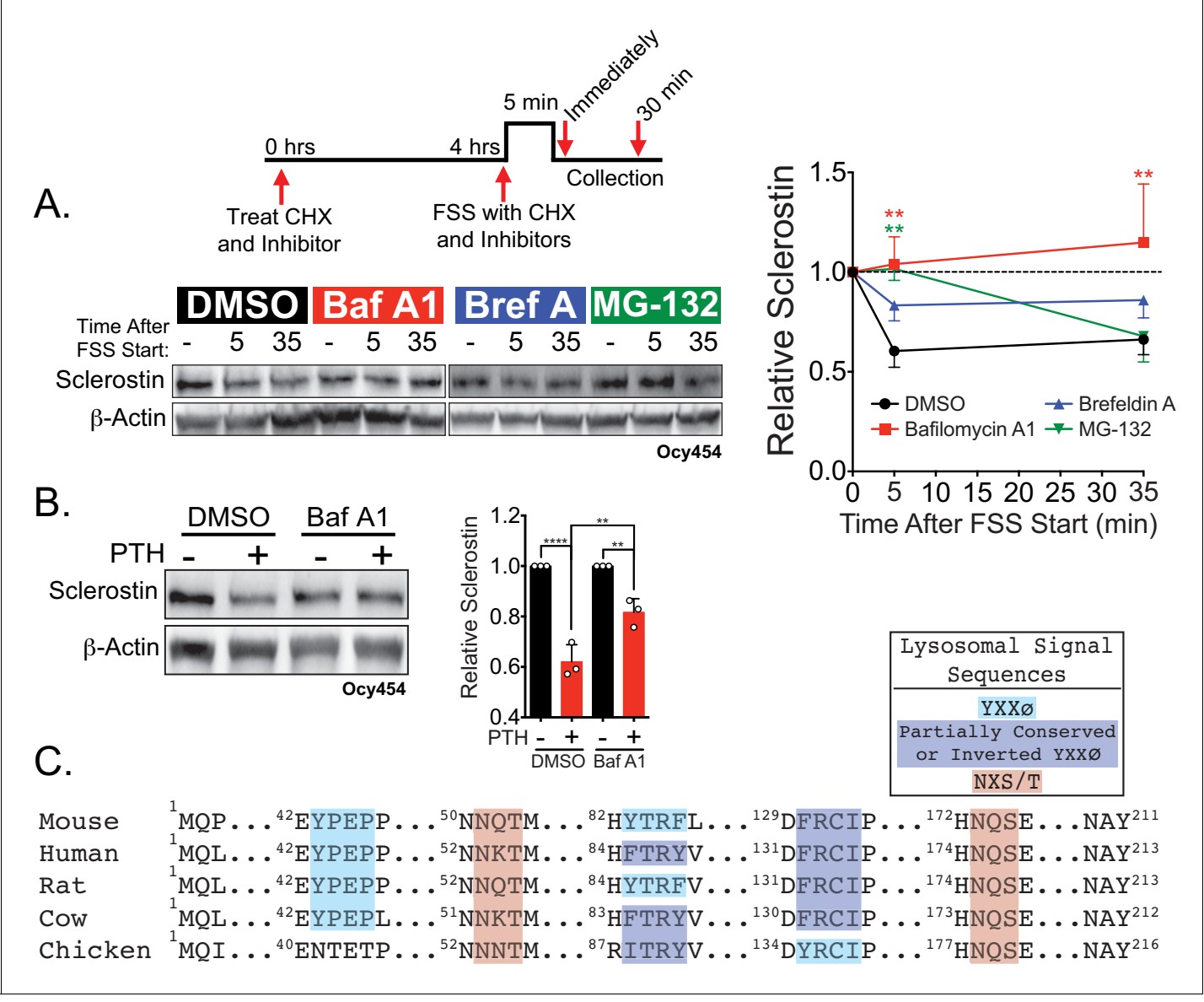

**Figure 3.** Sclerostin is rapidly degraded by the lysosome following bone anabolic stimuli. (A) Ocy454 cells transfected with GFP-sclerostin were treated with cycloheximide (150 µg/mL) to prevent new protein synthesis and either DMSO (0.1%), bafilomycin A1 (100 nM) to inhibit lysosomal degradation, brefeldin A (2 µm) to inhibit secretion, or MG-132 (10 µm) to inhibit the proteasome 4 hr prior to FSS. Cells were subjected to 5 min of FSS at 4 dynes/cm$^2$ and lysed immediately after the end of FSS or 30 min after the conclusion of FSS. Western blots were probed for sclerostin and β-actin. Time courses show mean ± SEM (n = 3–6 independent experiments/group). (B) Ocy454 cells transfected with GFP-sclerostin were pre-treated with DMSO (0.1%) or bafilomycin A1 (100 nM) to inhibit lysosomal degradation for 30 min prior to the addition of vehicle or PTH (1–34) (10 nM) for an additional 30 min (n = 3). Sclerostin abundance relative to the loading control was quantified. Graph depicts mean ± SD. *p<0.05, **p<0.01, ****p<0.0001 by two-way ANOVA with Holm–Sidak post hoc correction. (C) Amino acid sequences for sclerostin from mouse, human, rat, cow, and chicken were aligned using NCBI COBALT. Abbreviated sequences are shown and are annotated for putative lysosomal signal sequences. Full sequences are presented in *Figure 3—figure supplement 1*.

The online version of this article includes the following figure supplement(s) for figure 3:

**Figure supplement 1.** Sclerostin is rapidly degraded by the lysosome.

fusion of secretory vesicles, directs vesicles to lysosomes, and controls secretory granules in insulin-secreting beta-cells (*Waselle et al., 2003*; *Fukuda, 2013*). This lysosome-associated Rab27a secretory pathway also regulates RANKL secretion in osteoblasts (*Kariya et al., 2011*), suggesting the

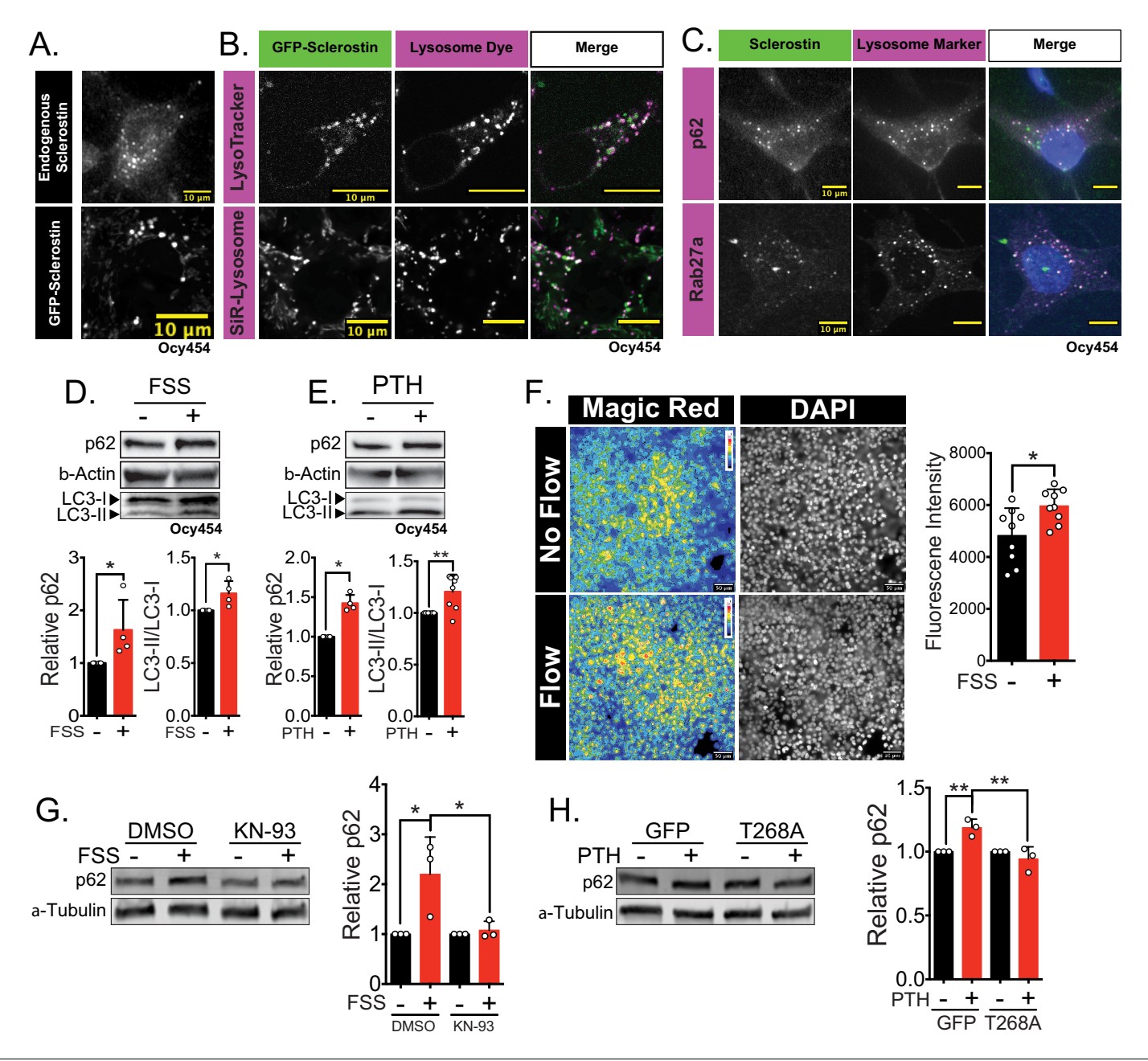

**Figure 4.** Sclerostin co-localizes with lysosomal markers, and both FSS and PTH induce lysosome activation through CaMKII. (A) Endogenous sclerostin (top) and GFP-tagged sclerostin (bottom) form discrete puncta in Ocy454 cells. (B) Ocy454 cells were transfected with GFP-sclerostin, and lysosomes were visualized with Lysotracker (1 mM, 1 hr) or siR-Lysosome (1 μM, 4 hr). Scale bar represents 10 μm. (C) Ocy454 cells were stained for endogenous sclerostin and either p62/sequestosome-1 or Rab27a to evaluate co-localization with these lysosome-associated proteins. (D) Ocy454 cells were exposed to 1 min of FSS at 4 dynes/cm$^2$, lysed immediately post-flow, and western blotted for p62/sequestosome-1, β-actin, and LC3 (n = 4). (E) Ocy454 cells were treated with PTH (1–34) (10 nM) for 5 min, lysed, and western blotted for p62/sequestosome-1 and β-actin (n = 4) and LC3 (n = 8). (F) UMR106 cells were subjected to FSS for 5 min, then Magic Red Cathepsin B was applied for 10 min, fixed, and imaged to assess lysosome activity (n = 9). (G) Ocy454 cells were treated with DMSO or KN-93 (10 μM) to inhibit CaMKII for 1 hr prior to FSS at 4 dynes/cm$^2$ for 5 min before lysing immediately after FSS. Western blots were probed for p62/sequestosome-1 and β-actin (n = 3). (H) Ocy454 cells were transfected with a plasmid expressing either GFP or dominant negative CaMKII T286A prior to treatment with PTH (1–34) (10 nM) for 30 min. Western blots were probed for p62/sequestosome-1 and β-actin (n = 3). Graphs depict mean ± SD. *p<0.05, **p<0.01 by unpaired two-tailed t-test (D–F) or by two-way ANOVA with Holm–Sidak post hoc correction (G, H).

The online version of this article includes the following figure supplement(s) for figure 4:

*Figure 4 continued on next page*

*Figure 4 continued*

**Figure supplement 1.** Co-localization of sclerostin with lysosomes.

presence of a common mechanism in osteoblast-lineage cells for controlling the abundance of secreted proteins that control bone remodeling.

## Lysosome activity is regulated by bone anabolic stimuli

Next, we examined if FSS or PTH altered lysosomal activity. Both FSS and exposure to PTH increased p62/sequestosome-1 protein abundance and increased LC3-II/LC3-I ratio, consistent with enhanced lysosomal delivery of autophagy cargo (*Figure 4D,E*). Similarly, FSS induced an increase in lysosome activity, as determined using Magic Red Cathepsin B activity assay (*Figure 4F*). CaMKII activation regulates lysosomal activity and protein degradation in other tissues (*Reventun et al., 2017*; *Li et al., 2017*; *Zemoura et al., 2019*; *Sandberg and Borg, 2006*). Given that both FSS and PTH activate CaMKII in osteocytes (*Lyons et al., 2017*; *Quinn et al., 2000*; *Williams et al., 2016*), we examined if FSS and PTH converge on CaMKII as a common integrator regulating lysosome activation. To do this, Ocy454 cells were treated with KN-93 to inhibit CaMKII or were transfected with a dominant negative CaMKII construct (T286A), and then exposed to FSS or PTH, respectively. In both CaMKII-disrupted cell populations, FSS and PTH failed to increase p62/sequestosome-1 protein abundance (*Figure 4G,H*), supporting that CaMKII activation following anabolic stimuli is necessary for lysosome activation. Together, these data support that osteocytes increase their lysosomal following bone anabolic stimuli and that CaMKII is likely a converging point between the FSS and PTH with respect to lysosomal activation.

## Nitric oxide is necessary and sufficient for rapid sclerostin degradation following FSS

This regulated lysosomal degradation of sclerostin following FSS or PTH treatment is reminiscent of a relatively obscure autophagy pathway known as crinophagy, which is utilized by peptide-secreting cells, such as pancreatic beta-cells, to route secretory vesicles to the lysosome rather than being delivered to the membrane for exocytosis (*Lee et al., 2019*; *Weckman et al., 2014*). Interestingly, crinophagy is regulated by nitric oxide (*Sandberg and Borg, 2006*). In osteocytes, nitric oxide production following mechanical cues is a canonical response (*Klein-Nulend et al., 1998*; *Zaman et al., 1999*; *Klein-Nulend et al., 2014*; *Nagayama et al., 2019*; *Gohin et al., 2016*); however, the direct biological consequence of nitric oxide on the potent activation of osteoblasts and bone formation has remained incomplete, though a link between *Sost* mRNA and sclerostin protein abundance and nitric oxide has been suggested, albeit on a 24 hr timescale (*Callewaert et al., 2010*). Accordingly, we examined if nitric oxide could also contribute to this crinophagy-like, rapid degradation of sclerostin protein in osteocytes.

Indeed, activation of nitric oxide signaling in Ocy454 cells with S-Nitroso-N-acetyl-DL-penicillamine (SNAP), a nitric oxide donor, induced a rapid decrease in sclerostin protein (*Figure 5A*). This rapid reduction of sclerostin protein occurred without increased CaMKII phosphorylation (*Figure 5A*), suggesting that activation of nitric oxide production may be downstream of CaMKII phosphorylation. When Ocy454 cells transfected with myc-sclerostin were treated with SNAP, a nitric oxide donor, in the presence of the lysosome inhibitor bafilomycin A1, sclerostin degradation was prevented (*Figure 5B*), confirming that nitric oxide is sufficient to drive the lysosomal degradation of sclerostin. Additionally, blocking nitric oxide production with L-NAME prevented the FSS-activated degradation of sclerostin protein at 5 min, as well as the FSS-induced increase in p62/sequestosome-1 (*Figure 5C*), supporting a role of nitric oxide in the activation of the lysosome and degradation of sclerostin, likely upstream of the lysosome and downstream of CaMKII. In total, these data support that sclerostin is directed through a defined secretory pathway unless directed to the lysosome by mechano-transduction or PTH-initiated signaling.

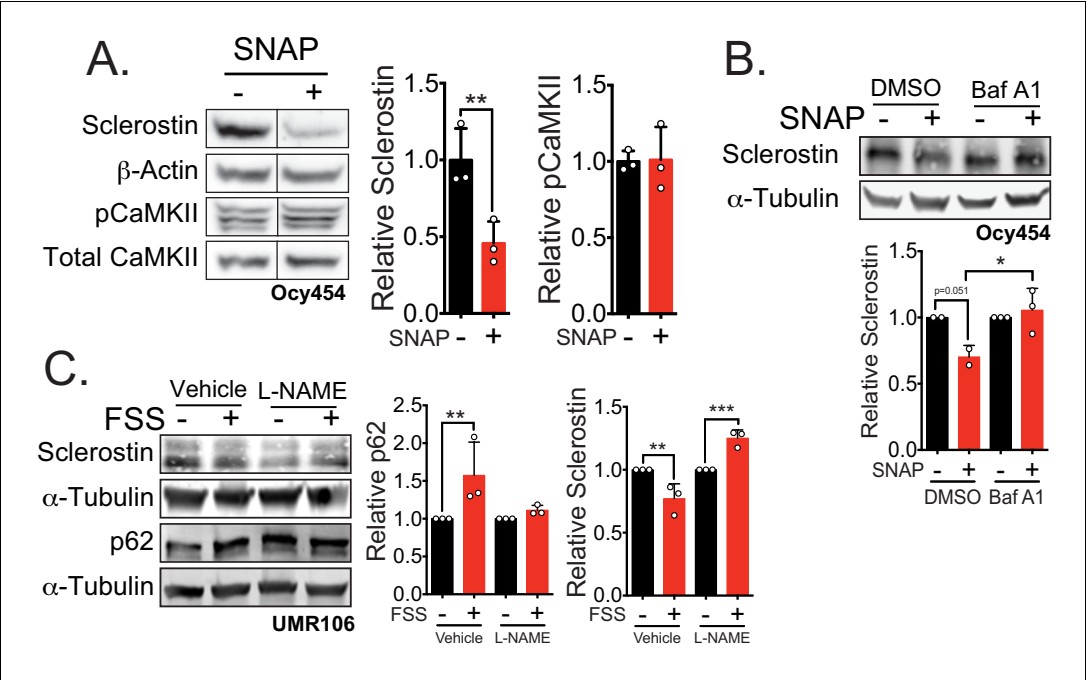

**Figure 5.** Nitric oxide, known mechanical load effector in bone and a molecular controller of crinophagy, contributes to sclerostin degradation. (**A**) Ocy454 cells transfected with GFP-sclerostin were treated with vehicle (water) or 10 μM SNAP, a nitric oxide donor, and lysed after 5 min. Western blots were probed for sclerostin, α-tubulin, pCaMKII, and total CaMKII (n = 3). For each antibody, blots are from a single gel and exposure; a vertical black line indicates removal of irrelevant lanes. (**B**) Ocy454 cells transfected with myc-tagged sclerostin were treated with DMSO or bafilomycin A1 (100 nM) to inhibit lysosomal degradation, for 30 min, then treated with SNAP, a nitric oxide donor, for 5 min and lysed. Western blots were probed for sclerostin and α-tubulin (n = 2–3). (**C**) UMR106 cells were treated with vehicle or L-NAME (1 mM) to inhibit nitric oxide synthases (NOSs) for 1 hr and then exposed to 1 or 5 min of FSS. Lysates from cells exposed to 1 min of FSS were probed for p62/sequestosome-1 and α-tubulin abundance and lysates from cells exposed to 5 min of sclerostin were probed for sclerostin and α-tubulin abundance (n = 3). Graphs depict mean ± SD. *p<0.05, **p<0.01, ***p<0.001 by unpaired two-tailed t-test (**A**) or two-way ANOVA with Holm–Sidak post hoc test (**B**, **C**).

## NOX2 ROS and lysosome activity are necessary for sclerostin degradation and bone formation following mechanical stress in vivo

To translate our findings to in vivo models, we examined the contribution of both the upstream mechano-transduction cascade originally described in vitro (*Lyons et al., 2017*) and the lysosome to load-induced sclerostin degradation and bone formation. First, we tested the effects of targeting the upstream mechano-transduction pathway that converges on sclerostin degradation on in vivo bone formation following mechanical load. Our prior work revealed that NOX2-derived ROS is an essential early step in the mechano-transduction pathway converging on sclerostin protein loss (*Lyons et al., 2017*). Both hydrogen peroxide (*Lyons et al., 2017*) or ROS generated by a genetically encoded photoactivatable protein, KillerRed, were sufficient to activate CaMKII and decrease sclerostin protein abundance in Ocy454 and UMR106 cells, respectively (*Figure 6—figure supplement 1A,B*). To validate the fidelity and conservation of this ROS-mediated sclerostin degradation in an intact bone, we treated tibiae with hydrogen peroxide ex vivo to mimic mechanical stress. Like results found in vitro, sclerostin protein was rapidly decreased in tibiae treated with hydrogen peroxide, supporting a role of ROS in the control of sclerostin abundance in intact bone (*Figure 6A*).

We next examined how NOX2 contributes to load-induced bone formation in vivo. We performed ulnar loading on 13 week old male mice pre-treated with vehicle or the NOX2 inhibitor apocynin 2 hr prior to load (*Figure 6B*). Apocynin pre-treatment and ulnar loading were repeated once a day for four consecutive days. After the final day of loading (day 4), inhibitor treatments were ceased, the bone surfaces were subsequently labeled with alizarin red (day 4) and calcein (day 11), and

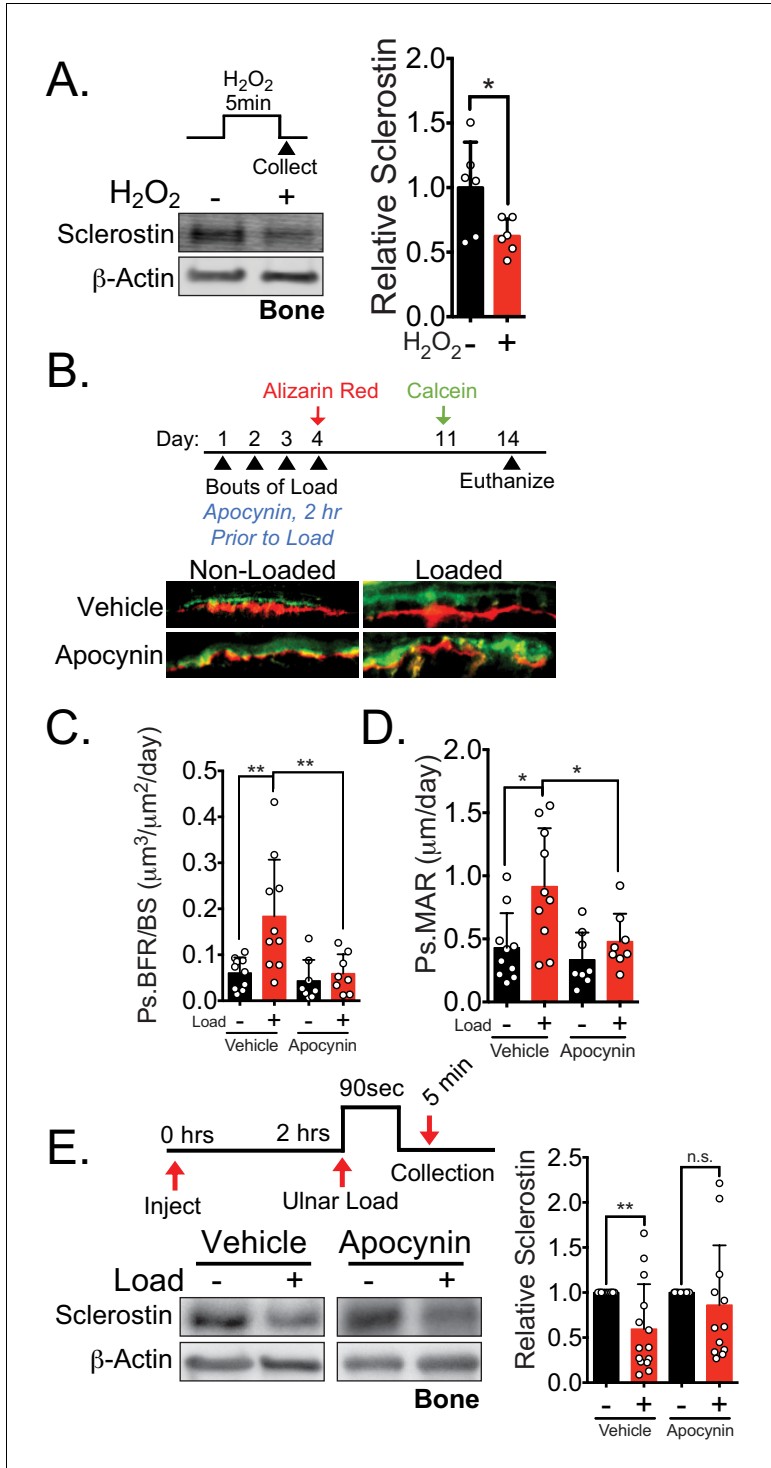

**Figure 6.** NOX2-dependent ROS are necessary for load-induced sclerostin degradation and bone formation in vivo. (**A**) Dissected ulnae and radii flushed of marrow were treated with hydrogen peroxide (100 μM) as a source of ROS for 5 min before homogenization. Western blots were probed with sclerostin and β-actin (n = 6 mice). (**B**) Thirteen week old male C57Bl/6 mice treated with vehicle (saline, n = 10 mice) or apocynin (3 mg/kg, n = 8 mice) to inhibit NOX2 were forearm loaded (1800 με, 90 s, 2 Hz) and labeled with calcein and alizarin red at the indicated times for dynamic histomorphometry. Representative periosteal double labeling is shown. (**C**) Periosteal bone formation rate (Ps.BFR) and (**D**) periosteal mineral apposition rate (Ps.MAR) were calculated. (**E**) Fourteen to 17 week old male and female C57Bl/6 mice treated with vehicle (saline + 4% DMSO, i.p., n = 14) or apocynin (3 mg/kg in saline, i.p., n = 12 mice) to inhibit NOX2 were treated 2 hr prior to ulnar loading (2000 με, 90 s, 2 Hz).
*Figure 6 continued on next page*

*Figure 6 continued*

Non-loaded and loaded limbs were isolated 5 min post-load, and western blots were probed for sclerostin and β-actin. Vehicle data is duplicated in *Figure 7D* as all animals were run and processed together. Graphs depict mean ± SD. *p<0.05, **p<0.01 by unpaired two-tailed t-test (**A**), two-way ANOVA with Holm–Sidak post hoc correction (**C**, **D**), or Kruskal–Wallis with Dunn's post hoc correction (**E**).

The online version of this article includes the following figure supplement(s) for figure 6:

**Figure supplement 1.** ROS is sufficient to drive CaMKII activation and loss of sclerostin protein.

dynamic histomorphometry was performed to assess de novo bone formation. Acute administration of apocynin 2 hr prior each bout of ulnar loading blocked the subsequent load-induced increase in periosteal bone formation rate (Ps.BFR) (*Figure 6B,C*) and mineral apposition rate (Ps.MAR) (*Figure 6B,D*) observed 14 days after the initiation of the experiment. To confirm that in apocynin treated animals failed to degrade sclerostin following ulnar loading, sclerostin protein abundance was assessed in 14–17 week old male and female mice pretreated with vehicle or apocynin to inhibit NOX2 subjected to a single bout of ulnar load. Ulnae were harvested 5 min post-load, a time frame consistent with in vitro and in vivo observations, and osteocyte-enriched cortical bone lysates were profiled by western blotting. Inhibition of NOX2 with apocynin blunted the rapid degradation of sclerostin protein in vivo (*Figure 6E*), supporting that NOX2 ROS is necessary for sclerostin degradation.

Second, to compliment the NOX2 targeting data in vivo, we next wanted to examine the impact of targeting lysosomal degradation on load-induced bone formation and sclerostin degradation in vivo. Ulnar loading was performed on 15 week old male mice treated with vehicle or the lysosome inhibitor bafilomycin A1 once a day for four consecutive days (*Figure 7A*). After the final day of loading, inhibitor treatment was ceased, the bone surfaces were labeled with alizarin red and calcein, and dynamic histomorphometry was performed to assess de novo bone formation. As for apocynin, bafilomycin A1 was not administered during the inter-label period when bone formation was monitored. Acute administration of bafilomycin A1 4 hr prior to each bout of ulnar load reduced the subsequent load-induced increase in Ps.BFR (*Figure 7A,B*) and Ps.MAR (*Figure 7A,C*) observed 14 days after the initiation of the experiment. In contrast, the administration of bafilomycin A1 during the first four days of the experiment had no effect on the Ps.BFR or Ps.MAR in the contralateral non-loaded limb. To validate the effect of lysosomal inhibition on sclerostin degradation, we treated animals acutely with bafilomycin A1 prior to a single bout of ulnar loading to examine sclerostin protein abundance. Ulnae were harvested 5 min post-load, a time frame consistent with in vitro and in vivo observations, and osteocyte-enriched cortical bone lysates were profiled by western blotting. Inhibition of lysosomal degradation with bafilomycin A1 blunted the rapid degradation of sclerostin protein in vivo (*Figure 7D*), supporting that lysosomal function is necessary for the rapid degradation of sclerostin.

When interpreted together, these results demonstrate that blocking the upstream mechano-pathway by targeting NOX2 with apocynin or targeting the final lysosomal degradation of sclerostin with bafilomycin A1 both prevented load-induced sclerostin degradation and subsequent bone formation. These data strongly support the relevance of post-translational control of sclerostin to skeletal physiology and adaptation to mechanical loading in vivo.

## Disrupted lysosomal function in Gaucher disease leads to sclerostin dysregulation

Having defined sclerostin regulation by the lysosome, we probed the clinical relevance using induced pluripotent stem cell (iPSC)-derived osteoblasts from Gaucher disease, a lysosomal storage disorder in which patients lack the lysosomal hydrolase glucocerebrosidase (GCase). As with many lysosomal storage disorders, patients with Gaucher disease exhibit skeletal dysplasias and low bone mass (*Hughes et al., 2019*). Gaucher disease iPSC-derived osteoblasts display defective osteoblast differentiation and mineralization caused by defects in Wnt/β-catenin signaling, an effect that is reversed by treatment with recombinant GCase (*Panicker et al., 2018*). We speculated that this suppression of β-catenin signaling, osteoblast differentiation, and osteoblast function may be a consequence of defects in sclerostin control, as sclerostin inhibits these processes. Consistent with

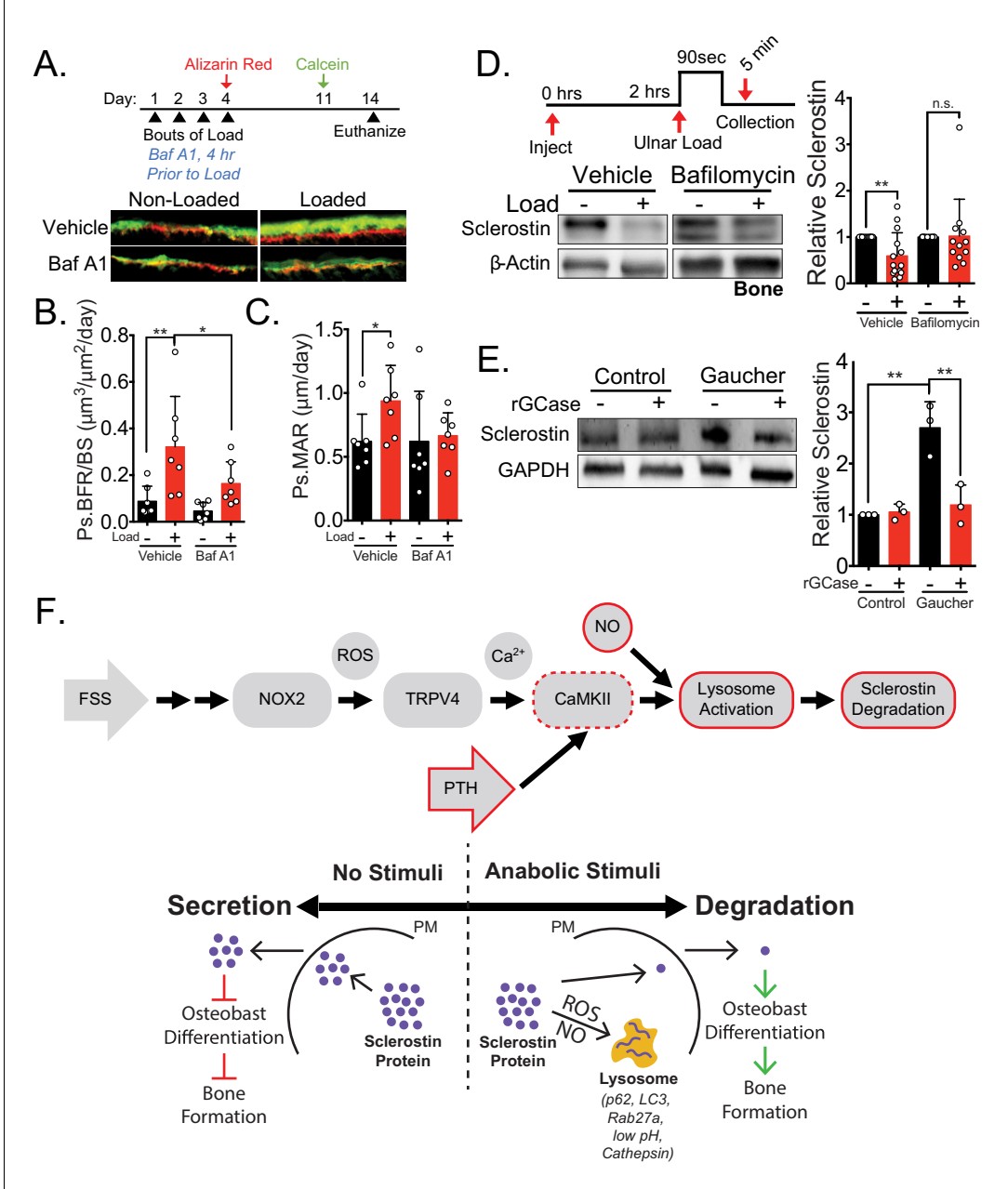

**Figure 7.** Lysosomal function is necessary for load-induced sclerostin degradation and subsequent bone formation and is implicated in human disease. (A) Fifteen week old male C57Bl/6 mice treated with vehicle (saline + 4% DMSO, n = 7 mice) or bafilomycin A1 (1 mg/kg, n = 7 mice) to inhibit lysosomal degradation were forearm loaded (2000 με, 90 s, 2 Hz) and labeled with calcein and alizarin red at the indicated times for dynamic histomorphometry. Representative periosteal double labeling are shown. (B) Periosteal bone formation rate (Ps.BFR) and (C) periosteal mineral apposition rate (Ps.MAR) were calculated. (D) Fourteen to 17 week old male and female C57Bl/6 mice treated with vehicle (saline + 4% DMSO, i.p., n = 14) or bafilomycin A1 (1 mg/kg in saline + 4% DMSO, i.p., n = 12 mice) to inhibit lysosomal degradation were treated 2 hr prior to ulnar loading (2000 με, 90 s, 2 Hz). Non-loaded and loaded limbs were isolated 5 min post-load, and western blots were probed for sclerostin and β-actin. Vehicle data is duplicated in *Figure 6E* as all animals were run and processed together. (E) Human iPSC-derived osteoblasts from either control (non-diseased) or Gaucher disease patients were treated with vehicle or recombinant glucocerebrosidase (rGCase, 0.24 U/mL) for 5 days, then lysed for western blotting. Western blots were probed for sclerostin and GAPDH (n = 3 independent patient-derived iPSC lines/group). Graphs depict mean ± SD. *p<0.05, **p<0.01 by two-way ANOVA with Holm–Sidak post hoc correction (B, C, and E) or Kruskal–Wallis with Dunn's post hoc correction (D). (F) FSS causes the rapid degradation of sclerostin protein by the lysosome through a number of molecular mediators. PTH, converging with this FSS mechano-transduction pathway at CaMKII, also mediates the lysosomal degradation of sclerostin protein. Icons outlined red are molecular mechanisms controlling sclerostin abundance that have been described within this manuscript that integrate into our previously described mechano-transduction

*Figure 7 continued on next page*

*Figure 7 continued*

cascade. Osteoanabolic stimuli, working through reactive oxygen (ROS) and reactive nitrogen species (RNS), direct sclerostin to the lysosome for degradation. This results in reduced sclerostin to allow for bone formation. PM: plasma membrane; ROS: reactive oxygen species; NO: nitric oxide.

lysosomal degradation of sclerostin, we observed that iPSC-derived osteoblasts from Gaucher patients had significantly increased levels of sclerostin compared to iPSC-derived osteoblasts from healthy patients without Gaucher disease (*Figure 7E*). Furthermore, treating Gaucher iPSC-derived osteoblasts with recombinant GCase, which restores lysosomal function, β-catenin signaling, and osteoblast differentiation (*Panicker et al., 2018*), also restored sclerostin abundance to control levels (*Figure 7E*). Since sclerostin acts as an inhibitor of Wnt/β-catenin signaling, these data suggest sclerostin as a therapeutic target for bone loss in Gaucher disease.

## Discussion

Our data show that osteocytes respond to distinct osteoanabolic cues, mechanical load and PTH, by redirecting sclerostin from a secretory pathway to the lysosome for rapid degradation (*Figure 7F*). In response to mechanical load, this signaling event is mediated by activation of ROS and nitric oxide signaling, a CaMKII-dependent increase in lysosomal activity, and the post-translational degradation of sclerostin by the lysosome. For PTH, CaMKII is likely a common integrator upstream of lysosomal degradation of sclerostin protein. Furthermore, we demonstrate that both lysosomal activity and activation of the upstream of mechano-signaling pathway are required for mechanically induced decreases in sclerostin protein and bone formation in vivo. These findings inform a new model of osteocyte mechano-transduction and shuttling of sclerostin to lysosomes that integrates many of the molecular effectors of the osteocyte mechano-response, including calcium, the cytoskeleton, nitric oxide, and sclerostin (*Thompson et al., 2012*; *Schaffler et al., 2014*; *Geoghegan et al., 2019*; *Baik et al., 2013*) and unifies many important discoveries related to osteoanabolic signals that act through sclerostin. These findings also describe a role of nitric oxide production, a canonical response to mechanical load in bone of previously unclear functional consequence, in the regulated degradation of sclerostin protein. Finally, we link sclerostin degradation not only to skeletal physiology in mice but also to human disease using Gaucher disease iPSCs.

These data do not preclude a transcriptional control of the *Sost* gene by osteoanabolic stimuli (*Tu et al., 2012*; *Sato et al., 2020*). Rather, they add an important downstream check point for regulating sclerostin protein bioavailability with remarkable temporal control. It may be that post-translational control is the first line response to a bone anabolic stimulus, whereas long-term stimulation leads to genome level transcriptional control of the *Sost* gene. Indeed, longer periods of loading are required to observe *Sost* mRNA decreases than are required for loss of protein described here. Our brief, 5 min FSS (*Williams et al., 2020*) and short, 30 min duration PTH exposure (*Figure 2—figure supplement 1*) are insufficient to decrease *Sost* mRNA abundance hours after the stimulus. However, the contribution of longer duration stimulation of this NOX2-Ca$^{2+}$-CaMKII pathway to *Sost* mRNA is not yet known.

In isolation, the effects of inhibiting either NOX2-dependent ROS or lysosomal function on load-induced bone formation and sclerostin abundance could be influenced by the broad pharmacological impact of these drugs. However, that two distinct pharmacological inhibitors targeting different steps in the pathway controlling sclerostin yielded a similar effect on bone formation and sclerostin protein abundance following load supports our conclusions. This conclusion is strengthened by two other important considerations: firstly, there was also no statistical effect of either apocynin or bafilomycin A1 on bone formation rate or mineral apposition rate in the contralateral, non-loaded limb. Secondly, the inhibitors were only present for each bout of ulnar load but were absent during the 7 day bone forming period, indicating that disruption of the acute downstream response to the mechanical cue, not the ability of the osteoblasts to build bone, caused the reductions in bone formation.

The rapid, controlled degradation of osteocyte sclerostin has intriguing parallels to crinophagy, a specialized autophagic process in peptide-secreting cells in which cellular cues elicit the re-routing of secretory proteins to the lysosome (*Lee et al., 2019*). Crinophagy is well described in pancreatic beta-cells, in which Rab27a-positive, insulin-filled secretory vesicles can be shuttled to the lysosome

for degradation when not needed, an effect triggered by nitric oxide (*Waselle et al., 2003*; *Sandberg and Borg, 2006*). Rab27a is associated with secretory granules and other lysosome-associated vesicles (*Fukuda, 2013*; *Tolmachova et al., 2004*), and, given the localization of nearly all the endogenous sclerostin with Rab27a, suggests sclerostin is packaged in vesicles. Thus, analogous to crinophagy, our data support that sclerostin is directed through a Rab27a-positive secretory pathway for exocytosis or is trafficked to the lysosome for degradation, an effect that can be regulated by CaMKII and mechano-activated nitric oxide. Notably, this Rab27a-associated secretory pathway functions in osteoblasts to control RANKL secretion (*Kariya et al., 2011*). Whether this mechano-pathway and regulated degradation extends to osteocyte-derived RANKL was not analyzed in the present study, but may be physiologically significant as osteocytes are the primary source of bioactive RANKL directing bone resorption in response to physiologic cues (*Xiong et al., 2015*; *Xiong et al., 2014*).

Using Gaucher disease as a model system, this work links lysosomal function to sclerostin regulation in disease. Gaucher patients experience many skeletal dysplasias, including low bone mass and osteoporosis. Though this cell autonomous defect in Gaucher osteoblasts has previously been linked to dysregulated β-catenin that impaired osteogenic differentiation and mineralization capacity (*Panicker et al., 2018*), the present findings reveal a possible mechanism by which β-catenin is suppressed. Sclerostin is a canonical Wnt/β-catenin antagonist; therefore, the increased sclerostin levels observed here in Gaucher disease iPSCs are likely to lead to the previously reported decrease in β-catenin activation and osteoblast differentiation (*Panicker et al., 2018*). Likewise, we showed that restoring expression of the missing hydrolase, glucocerebrosidase, in Gaucher disease iPSCs lowers sclerostin abundance, which could explain the rescue in β-catenin activation and osteoblast differentiation reported previously (*Panicker et al., 2018*). Our data support the exploration of therapeutics that target sclerostin to treat low bone mass symptoms in patients with lysosomal storage disorders.

That sclerostin protein abundance is post-translationally controlled may have important implications in age-related osteoporosis and the reduced sensitivity of osteocytes to mechanical cues with aging (*Hemmatian et al., 2017*; *Haffner-Luntzer et al., 2016*). Lysosome activity is diminished with age (*Moore, 2020*; *Stead et al., 2019*), including in the osteocyte during age-related bone loss (*Chen et al., 2014*), an effect that could impair sclerostin degradation and new bone formation even in the face of osteoanabolic signals. While undoubtedly multifaceted in its impacts, targeting autophagy and lysosome activity to improve bone mass in aging has been proposed (*Li et al., 2020*; *Wang et al., 2019*), and these data suggest that impacts on sclerostin bioavailability might contribute mechanistically to its efficacy.

Together, these discoveries provide key insights into the unexpected, rapid regulation of osteocyte sclerostin protein by the lysosome and reveal new therapeutic targets that can be exploited to improve bone mass in conditions such as osteoporosis. Additionally, targeting sclerostin may be important to interventions to improve bone mass in lysosomal storage disorders, like Gaucher disease.

# Materials and methods

**Key resources table**

| Reagent type (species) or resource | Designation | Source or reference | Identifiers | Additional information |
|---|---|---|---|---|
| Gene (*Mus musculus*) | Sost | National Center for Biotechnology Information | AAK13455 | |
| Gene (*Homo sapiens*) | Sost | National Center for Biotechnology Information | AAK16158.1 | |
| Gene (*Rattus norvegicus*) | Sost | National Center for Biotechnology Information | EDM06161.1 | |
| Gene (*Bos Taurus*) | Sost | National Center for Biotechnology Information | NP_001159986.1 | |

*Continued on next page*

*Continued*

| Reagent type (species) or resource | Designation | Source or reference | Identifiers | Additional information |
|---|---|---|---|---|
| Strain, strain background (*Mus musculus*, male and female) | C57Bl/6 | Jackson Laboratories | 000664 | RRID:IMSR_JAX:000664 |
| Cell line (*Mus musculus*) | Ocy454 | P. Divieti-Pajevic, Boston University | | RRID:CVCL_UW31 |
| Cell line (*Rattus norvegicus*) | UMR106 | ATCC | CRL-1661 | RRID:CVCL_3617 |
| Transfected construct (*Mus musculus*) | CaMKII T286A | Addgene | #29430 | |
| Transfected construct (*Homo sapiens*) | GFP-tagged sclerostin | Origene | #RG217648 | |
| Transfected construct (*Mus musculus*) | myc-tagged sclerostin | Origene | #MR222588 | |
| Transfected construct (*Mus musculus*) | KillerRed | Evrogen | FP966 | |
| Biological sample (*Homo sapiens*) | WT and GD iPSCs | PMID:23071332 | | |
| Antibody | Goat Polyclonal Anti-sclerostin | R and D Systems | AF1589 RRID:AB_2270997 | (1:250–500) |
| Antibody | Mouse Monoclonal Anti-GAPDH | Millipore | MAB374 RRID:AB_2107445 | (1:2500) |
| Antibody | Mouse Monoclonal Anti-βActin | Sigma | A1978 RRID:AB_476692 | (1:5000) |
| Antibody | Mouse Monoclonal Anti-αTubulin | Sigma | T9026 RRID:AB_477593 | (1:2000) |
| Antibody | Rabbit Polyclonal Anti-Col1a1 | Sigma | ABT257 RRID:AB_2890134 | (1:1000) |
| Antibody | Rabbit Monoclonal Anti-pCaMKII | Cell Signalling Technology | 12716S RRID:AB_2713889 | (1:1000) |
| Antibody | Rabbit Polyclonal Anti-Total CaMKII | Cell Signalling Technology | 3362S RRID:AB_2067938 | (1:1000) |
| Antibody | Rabbit Monoclonal Anti-p62/Sequestosome-1 | Cell Signalling Technology | 23214 RRID:AB_2798858 | (1:250–500) |
| Antibody | Rabbit Polyclonal Anti-LC3B | Cell Signalling Technology | 2775 RRID:AB_915950 | (1:500) |
| Antibody | Rabbit Monoclonal Anti-Rab27A | Cell Signalling Technology | 69295 RRID:AB_2799759 | (1:250) |
| Antibody | HRP Anti-Rabbit | Cell Signalling Technology | 7074 RRID:AB_2099233 | (1:1000–5000) |
| Antibody | HRP Anti-Mouse | Cell Signalling Technology | 7076 RRID:AB_330924 | (1:1000–5000) |
| Antibody | HRP Anti-Goat | Thermo Fisher Scientific | A27014 RRID:AB_2536079 | (1:1000) |
| Antibody | Donkey anti-Goat IgG Alexa Fluor 546 | Thermo Fisher Scientific | A-11056 RRID:AB_142628 | (1:100) |
| Antibody | Chicken anti-Rabbit Alexa Fluor 488 | Thermo Fisher Scientific | A-21441 RRID:AB_2535859 | (1:100) |
| Sequence-based reagent | *Sost* | NCBI BLAST | NM_024449.6 | GGA ATG ATG CCA CAG AGG TCA T and CCC GGT TCA TGG TCT GGT T |
| Sequence-based reagent | *Gapdh* | NCBI BLAST | NG_007785.2 | CGT GTT CCT ACC CCC AAT GT and TGT CAT CAT ACT TGG CAG GTT TCT |

*Continued on next page*

*Continued*

| Reagent type (species) or resource | Designation | Source or reference | Identifiers | Additional information |
|---|---|---|---|---|
| Sequence-based reagent | *Hprt* | NCBI BLAST | NM_013556.2 | AGC AGT ACA GCC CCA AAA TGG and AAC AAA GTC TGG CCT GTA TCC AA |
| Sequence-based reagent | *Rpl13* | NCBI BLAST | NM_016738.5 | CGA AAC AAG TCC ACG GAG TCA and GAG CTT GGA GCG GTA CTC CTT |
| Commercial assay or kit | Magic Red | Sigma | CS0370 | |
| Commercial assay or kit | siR-Lysosome | Spirochrome | SC016 | |
| Commercial assay or kit | Lysotracker | Thermo Fisher Scientific | L7528 | |
| Commercial assay or kit | High-Capacity RNA-to-cDNA Kit | Thermo Fisher Scientific | 4388950 | |
| Commercial assay or kit | Maxima SYBR Green/ROX qPCR Master Mix | Thermo Fisher Scientific | FERK0221 | |
| Commercial assay or kit | JetPrime Transfection kit | PolyPlus Transfection | 114–75 | |
| Chemical compound, drug | Bafilomycin A1 (in vitro studies) | Cell Signalling Technology | 54645 | |
| Chemical compound, drug | Bafilomycin A1 (in vivo studies) | Research Products International | 88899-55-2 | |
| Chemical compound, drug | Brefeldin A | Cell Signalling Technology | 9972 | |
| Chemical compound, drug | MG-132 | Cell Signalling Technology | 2194 | |
| Chemical compound, drug | Cycloheximide | Cell Signalling Technology | 2112 | |
| Chemical compound, drug | PTH (1–34) | US Biological Life Sciences | #P3109-24D | |
| Chemical compound, drug | Leupeptin | Millipore | EI8 | |
| Chemical compound, drug | Apocynin | Sigma | 178385 | |
| Chemical compound, drug | L-NAME | Millipore | N5751 | |
| Chemical compound, drug | SNAP | Sigma | N3398 | |
| Chemical compound, drug | Alizarin Red | Sigma | A3882 | |
| Chemical compound, drug | KN-93 | Sigma | K1358 | |
| Chemical compound, drug | Calcein | Sigma | C0875 | |
| Chemical compound, drug | Halt Protease and Phosphatase Inhibitor Cocktail (EDTA-free) | Thermo Fisher Scientific | 78440 | |
| Chemical compound, drug | SuperBlockPBS | Thermo Fisher Scientific | 37515 | |
| Software, algorithm | FIJI | ImageJ | RRID:SCR_002285 | |
| Software, algorithm | Nikon NIS Elements 5.2 | Nikon | RRID:SCR_014329 | |
| Other | REVERT Total Protein Stain | Licor | 827–15733 | |
| Other | Prolong Gold Antifade Reagent with DAPI | Cell Signalling Technology | 8961S | |

*Continued*

| Reagent type (species) or resource | Designation | Source or reference | Identifiers | Additional information |
|---|---|---|---|---|
| Other | TRIzol | Sigma | T9424 | |
| Other | DCF | Invitrogen | D399 | |

## Chemicals and reagents

Bafilomycin A1 (in vitro studies, #54645), MG-132 (#2194), brefeldin A (#9972), cycloheximide (#2112), and antibodies against Thr 286 pCaMKII (#12716S), total CaMKII (3362S), p62/sequestosome-1 (#23214), LC3B (#2775), Rab27a (#69295), and Prolong Gold Antifade Reagent with DAPI (#8961S) were from Cell Signaling Technologies. REVERT Total Protein Stain (827–15733) was from Licor. Anti-sclerostin antibodies (#AF1589) were purchased from R and D Systems. Bafilomycin A1 (in vivo studies, #88899-55-2) was from Research Products International. Leupeptin (#EI8), $N_\omega$-nitro-L-arginine methyl ester hydrochloride (L-NAME, N5751), and GAPDH (MAB374) were from Millipore. PTH (1–34, #P3109-24D) was from US Biological Life Sciences. Alizarin red (#A3882), calcein (#C0875), Apocynin (178385), S-Nitroso-N-acetyl-DL-penicillamine (SNAP, N3398), and antibodies against β-actin (A1978), α-tubulin (T9026), Pro-Collagen Type I, A1 (Col1α1) (ABT257), Magic Red Cathepsin B Detection Assay Kit (CS0370), KN-93 (K1385), and TRIzol (T9424) were from Sigma. siR-Lysosome (CY-SC016) was from Spirochrome. GFP-tagged human sclerostin (#RG217648)- and myc-tagged mouse sclerostin (#MR222588) were purchased from Origene. KillerRed plasmid (FP966) was purchased from Evrogen. CaMKII T286A dominant negative construct was from Addgene (#29430). Recombinant human GCase (rGCase) (Cerezyme) was obtained from patient infusion remnants. 2′,7′-Dichlorofluorescein (DCF, D399) was purchased from Invitrogen. Halt Protease and Phosphatase Inhibitor Cocktail (EDTA-free) (78440), Lysotracker (L7528), SuperBlockPBS (37515), Donkey anti-Goat IgG Alexa Fluor 546 (A-11056), Chicken anti-Rabbit Alexa Fluor 488 (A-21441), High-Capacity RNA-to-cDNA Kit (4388950), and Maxima SYBR Green/ROX qPCR Master Mix (FERK0221) were from Thermo Fisher Scientific. JetPrime Transfection kit (114-75) was from PolyPlus Transfection. Modified RIPA lysis buffer contained 50 mM Tris–HCl pH 8.0, 150 mM NaCl, 1.0% NP-40, 0.5% sodium deoxycholate, 0.1% SDS, 10 mM $Na_4P_2O_7$, 10 mM 2-glycerolphosphate, 10 mM NaF, 10 mM EDTA, 1 mM EGTA, 1× HALT phosphatase, and protease inhibitor cocktail.

## Cell culture

UMR106 cells (purchased from ATCC, CRL-1661) were cultured in Dulbecco's modified essential medium (DMEM) supplemented with 10% fetal bovine serum (FBS), and maintained at 37°C and 5% $CO_2$, as described (*Gupta et al., 2016*). Ocy454 cells (provided by P. Divieti-Pajevic, Boston University) were cultured on type I rat-tail collagen-coated plates in α-minimal essential medium (αMEM) supplemented with 10% FBS and maintained at 33°C and 5% $CO_2$ (*Wein et al., 2015*; *Spatz et al., 2015*). Cell phenotype was verified by expression of bone cell specific markers, most importantly the osteocyte specific expression of sclerostin. Cells were used at low passages (<15) to maintain phenotype and tested by PCR to ensure the absence of mycoplasma. Prior to experimentation, cells were seeded into tissue-culture-treated vessels and maintained overnight at 37°C and 5% $CO_2$. The iPSC from a patient with type 2 Gaucher disease and a control subject used in this study have been previously described (*Panicker et al., 2018*). Their genotypes are as follows: W184R/D409H and WT/WT (Control MJ). Control and Gaucher disease iPSC were differentiated to osteoblasts as described (*Panicker et al., 2018*). Briefly, embryoid bodies from WT and Gaucher iPSCs were transferred to 0.1% (w/v) gelatin-coated plates and cultured in MSC medium (high-glucose DMEM [Invitrogen], 20% FBS [Hyclone], 1 mM L-glutamine, and 100 U/ml Pen/Strep [Invitrogen]) to generate mesenchymal stem cells (MSCs). To generate osteoblasts, the WT and Gaucher MSCs were plated at a density of $2 \times 10^4$ cells/cm$^2$ and were cultured in osteoblast differentiation media (MSC media supplemented with 10 mM beta-glycerophosphate [Sigma], 100 µM dexamethasone [Sigma], and 50 µg/mL ascorbic acid [Sigma]) for 3–4 weeks, as described. Recombinant glucocerebrosidase (rGCase, 0.24 U/ml) was added to the cultures with each media change.

## Fluid flow

Ocy454 and UMR106 cells were exposed to fluid flow using a custom FSS device (*Lyons et al., 2017*; *Lyons et al., 2016*). Media was removed, and cells were rinsed in a Hepes-buffered Ringer solution containing 10 mM Hepes (pH 7.3), 140 mM NaCl, 4 mM KCl, 1 mM $MgSO_4$, 5 mM $NaHCO_3$, 10 mM glucose, and 1.8 mM $CaCl_2$. Ringer solution was also used as fluid flow buffer. Cells were exposed to 1–5 min of FSS (four dynes/$cm^2$), as indicated, and lysed in a modified RIPA buffer plus HALT protease and phosphatase inhibitors at the time indicated in each experiment.

## Cell treatments

To block cellular degradation pathways, cells were pre-treated with bafilomycin A1 (100 nM, 30 min or 4 hr, as indicated), brefeldin A (2 μM, 4 hr), MG-132 (10 μM, 4 hr), leupeptin (200 μM, 6 hr), or dimethyl sulfoxide (DMSO) (0.1%, 30 min or 4 hr, as indicated) in Ringer solution. For PTH treatment, cells were treated with PTH (1–34) (10 nM) in Ringer solution for up to 30 min. For SNAP treatment, cells remained in the media they were plated in and SNAP dissolved in sterile water was added to a final concentration of 10 μM. To block the lysosomal function before SNAP treatment, cells were treated with bafilomycin A1 (100 nM) for 30 min prior to the addition of SNAP. Cells were lysed 5 min after addition of SNAP. For L-NAME treatment, L-NAME was dissolved in sterile water and then diluted into Ringer solution (1 mM). UMR106 cells were pre-treated with L-NAME or vehicle control 1 hr prior to FSS, then exposed to 5 min of FSS, and lysed immediately after. Ocy454 cells were treated with 10 μM KN-93 for 1 hr prior to the addition of PTH (1–34) (10 nM) for an additional 10 min. To assess PTH effects on *Sost* mRNA, Ocy454 cells were sera starved in αMEM supplemented with 0.1% FBS overnight and were then treated with PTH (1–34) (10 nM) or vehicle for 30 min diluted in αMEM supplemented with 0.1% FBS at 37°C, 5% $CO_2$. Media was then switched for fresh in αMEM supplemented with 0.1% FBS, and cells were lysed in TRIzol 5.5 hr after.

## RT-qPCR

Cells lysed with TRIzol were processed using Direct-zol RNA Kit to isolate RNA according to manufacturer's instructions. RNA was reverse transcribed using High-Capacity RNA-to-cDNA Kit according to manufacturer's instructions. cDNA was used for RT-qPCR using Thermo Scientific Maxima SYBR Green/ROX qPCR Master Mix. The level of the *Sost* was simultaneously normalized to expression levels for *Gapdh*, *Rpl13*, and *Hprt*. PCR primers used are as follows: *Sost*: GGA ATG ATG CCA CAG AGG TCA T and CCC GGT TCA TGG TCT GGT T; *Rpl13*: CGA AAC AAG TCC ACG GAG TCA and GAG CTT GGA GCG GTA CTC CTT; *Gapdh*: CGT GTT CCT ACC CCC AAT GT and TGT CAT CAT ACT TGG CAG GTT TCT; and *Hprt*: AGC AGT ACA GCC CCA AAA TGG and AAC AAA GTC TGG CCT GTA TCC AA.

## Sequence alignment

Amino acid sequences were acquired from the National Center for Biotechnology Information's (NCBI) protein database. Accession numbers are as follows: mouse (AAK13455); human (AAK16158.1); rat (EDM06161.1); cow (NP_001159986.1); and chicken (XP_024999845.1). Sequence alignment was done using NCBI's Constraint-based Multiple Alignment Tool (COBALT). Lysosomal signal sequences (*Braulke and Bonifacino, 2009*) and the secretory signal peptide were annotated manually.

## Degradation assays

Ocy454 cells were treated with CHX (150 μg/mL), and either DMSO (0.1%), bafilomycin A1 (100 nM), brefeldin A (2 μM), or MG-132 (10 μM) diluted in supplemented αMEM for 4 hr or with leupeptin 200 μM for 6 hr at 37°C and 5% $CO_2$. Cells were then exposed to 5 min of FSS as described above in Ringer containing the appropriate treatment with CHX. Cells were lysed in a modified RIPA buffer + HALT protease and phosphatase inhibitors and collected for western blotting at 5 and 30 min post-flow. For basal degradation assays, UMR106 cells were treated with 150 μg/mL CHX for 0, 1, 2, or 4 hr and lysed. A best-fit linear line was constrained through y = 1 to determine protein half-life.

## Transient transfections

Ocy454 cells were seeded in 96-well plates at a density of 20,000 cells/well and incubated at 37°C for 24 hr. Transient transfections were performed using 0.025 µg/well GFP-tagged sclerostin, 0.05 µg/well Myc-tagged sclerostin, or 0.1 µg/well CaMKII T286A DNA mixed with 5 µL/well JetPrime buffer and 0.1 µL/well JetPrime reagent, as described (*Lyons et al., 2017*). After 16 hr of incubation at 37°C, the transfection medium was replaced with complete αMEM. The cells were incubated an additional 24 hr at 37°C prior to experiments. UMR106 cells for KillerRed experiments were plated on glass bottom 10 mm dishes at a density of 50,000 cells/dish. Transient transfections were performed using 2 µg/well KillerRed and 4 or 8 µL/well Jetprime reagent. Sixteen hours after incubation, transfection medium was replaced with complete DMEM and were incubated an additional 37°C prior to experiments.

## Fluorescence co-localization

Ocy454 cells were seeded at 10,000 cells/well and grown on a glass-bottom 96-well plate. Cells were transfected with GFP-Sclerostin as described above, where indicated. For immunofluorescence staining, cells were fixed with 1% paraformaldehyde, permeabilized with 0.1% Triton-X in PBS, and blocked with SuperBlockPBS for 1 hr, as described (*Kerr et al., 2015*). Primaries diluted in Super-BlockPBS against p62/sequestosome-1, Rab27a, and sclerostin were used at 1:250 and incubated overnight at 4C. Chicken anti-Rabbit 488 and Donkey anti-Goat 546 diluted in SuperBlockPBS were used at 1:100 and incubated for 3 hr at room temperature. Cells were mounted with ProLong Gold Antifade with DAPI and imaged with a Nikon Ti2 microscope with a SpectraX Light Engine and a ds-Qi2 Monochrome camera. For live cell imaging, Ocy454 cells were transfected with GFP-Sclerostin as described above. Lysosomes were labeled Lysotracker (1 µM) for 1 hr and then imaged with a Nikon C2 confocal microscope. For co-localization quantification, Ocy454 cells transfected with GFP-sclerostin were treated with siR-Lysosome (1 µM) for 4.5 hr at 37°C to label lysosomes. Cells were imaged on a Nikon C2 confocal microscope and Z-stacks with 0.5 µm steps were obtained in each well. These Z-stacks were then denoised using Nikon Denoise aI (NIS Elements 5.2). Co-localization in the denoised Z-stacks using the Mander's coefficients was measured using the FIJI plugin, JaCOP (*Bolte and Cordelières, 2006*).

## Magic red cathepsin B activity assay

UMR106 cells were exposed to FSS at 4 dynes/cm$^2$ for 5 min. Magic Red and Hoechst (1×) reagent were then added to all wells for 10 min. Cells were washed three times with warm 1× PBS and then fixed for 10 min with 1% paraformaldehyde. Cells were washed one time with 1× PBS and imaged at 20× with a Nikon Ti2 microscope with a SpectraX Light Engine and a ds-Qi2 Monochrome camera. Average Magic Red intensity from three images from three wells was measured by subtracting a mask image of the nuclei from the Magic Red signal. Magic Red signal was then measured on a per pixel basis, and average intensities were calculated.

## KillerRed imaging and protein isolation

For imaging, UMR106 cells transfected with KillerRed were loaded with DCF (10 µM, 30 min, 37°C) diluted in Ringer solution to track ROS production. After loading, cells were washed with fresh Ringer solution. Cells were stimulated with LED light for 40 s. DCF and KillerRed signals were imaged before and after exposure to LED light in the same cells. For western blotting, cells transfected with KillerRed were stimulated with LED light for 5 min and were lysed 5 min after the end of light exposure. No light controls were treated the same but not exposed to light.

## Animals

Male and female, age-matched C57BL/6 mice were purchased from Jackson Laboratory. Mice were group housed in micro-isolator cages, and food (standard rodent chow) and water were available ad libitum. Mice were maintained on a 12-h-light–12-h-dark cycle. Experiments were conducted on 13–16 week old mice, as indicated. All animal protocols were approved by the Animal care and Use Committee at the University of Maryland School of Medicine.

## Ex vivo treatments

Ulnae, radii, or tibiae were dissected from surrounding soft tissues, epiphyses were cut, and marrow was flushed with Ringers solution. Bones were acclimated in complete αMEM at 37°C and 5% $CO_2$ for at least 20 min before moving them into Ringers solution with appropriate treatments. For hydrogen peroxide experiments, bones were moved into Ringers solution containing vehicle control or 100 μM hydrogen peroxide for 5 min at 37°C. For PTH experiments, bones were moved into Ringers solution containing vehicle or 10 nM PTH (1–34) for 30 min. Bones were then removed from the Ringers solution and were homogenized in RIPA buffer + HALT protease and phosphatase inhibitor cocktail using a Bullet Blender (Next Advance), as described (*Buo et al., 2017*; *Moorer et al., 2017*). Extracts were subsequently used for western blotting analysis, as described below.

## In vivo loading

Bone strains generated by ulnar loading were calculated using a strain gauge (Micro Measurements #EA-06-015DJ-120) affixed with cyanoacrylate to the lateral, mid-diaphyseal aspect of ulnae of age- and sex-matched C57BL/6 mice. The upper limbs of euthanized mice were placed in a horizontal orientation in a uniaxial load device (Aurora Scientific, 305C-FP). Load was applied in a slow ramp at 0.05 N/s, and then load versus strain plots were calculated and used to determine strains at a given load, as described (*Melville et al., 2015*).

For acute isolation of protein, 14–17 week old male and female C57BL/6 mice were subjected to a single, acute bout of ulnar loading. Briefly, animals were either untreated (*Figure 1D*) or were treated with vehicle (saline with 4% DMSO, *Figures 6E* and *7D*), apocynin (3 mg/kg dissolved in saline, *Figure 6E*), or bafilomycin A1 (1 mg/kg, 4% DMSO in saline, *Figure 7D*) intraperitoneally 2 hr prior to ulnar loading. Mice were anesthetized (isoflurane), and their left upper limb was placed in a horizontal orientation in a uniaxial load device (Aurora Scientific, 305C-FP). A small pre-load was applied (0.4 N), and then the forearm was cyclically loaded with a sinusoidal wave at 2 Hz and a peak strain between 1800 and 2000 με, as specified, for 90 s, as described (*Lee et al., 2002*; *Tomlinson et al., 2017*). Five minutes following load, loaded and contralateral non-loaded (control) ulnae and radii were dissected from surrounding soft tissue, epiphyses removed, flushed of marrow, and homogenized in RIPA buffer + HALT protease and phosphatase inhibitor cocktail using a Bullet Blender (Next Advance), as described (*Buo et al., 2017*; *Moorer et al., 2017*). Extracts were subsequently used for western blotting analysis, as described below. For sclerostin immunofluorescence, ulnae were fixed in warm 10% buffered formalin overnight, sucrose embedded in 30% sucrose overnight at 4°C, and cryosectioned using the Kawamoto tape transfer method at 5 μm thickness (*Kawamoto and Kawamoto, 2014*). Sections contained both the non-loaded and loaded limbs. Sections were permeabilized and blocked in 0.04% saponin in SuperBlockPBS for 1 hr. 1:250 sclerostin primary was diluted in 1× PBS with 0.04% saponin for 36 hr. 1:200 Donkey anti-Goat 546 was incubated for 2 hr at room temperature before sections were mounted with ProLong Gold Antifade with DAPI and imaged with a Nikon Ti2 microscope with a SpectraX Light Engine and a ds-Qi2 Monochrome camera. All sections were imaged with the same laser strength and exposure time to ensure proper comparisons between sections. Sclerostin-positive osteocytes were counted using the Cell Counter plugin in FIJI and reported as a proportion of total osteocytes in a defined ROI.

For dynamic histomorphometry following mechanical stimulation, 13 week (apocynin experiments) or 15 week old (bafilomycin A1 experiments) male C57BL/6 mice were subjected to four consecutive days (days 1–4) of forearm loading at 1800 με (apocynin experiments) or 2000 με (bafilomycin A1 experiments) at 2 Hz for 90 s, as described above. For apocynin experiments, apocynin (3 mg/kg dissolved in saline) or vehicle control (saline) were injected intraperitoneally 2 hr prior to each bout of ulnar loading. The contralateral limb served as a non-loaded control in all loading experiments. Following loading, mice were returned to their cages for unrestricted activity. For bafilomycin A1 experiments, bafilomycin A1 (1 mg/kg, 4% DMSO in saline) or vehicle control (saline with 4% DMSO) was injected intraperitoneally 24 hr before the first bout of loading (day 0) and 4 hr prior to each bout of loading. For both inhibitors, treatment occurred only during the loading phase and not during the subsequent monitoring of bone formation by dynamic histomorphometry.

To assess cortical bone formation rate during the post-load period, animals were injected intraperitoneally with alizarin red (30 mg/kg) after the final bout of loading (day 4) and injected intraperitoneally with calcein (30 mg/kg) on day 11. Three days later (day 14), animals were euthanized,

loaded, and contralateral non-loaded ulnae and radii were dissected from surrounding soft tissue and stored in 100% ethanol. Bones were then placed in 30% sucrose in PBS at 4℃ overnight. Bones were embedded in optimal cutting temperature (OCT) compound and sectioned at 5 µm thickness using the Kawamoto film method (*Kawamoto and Kawamoto, 2014*). Three sections were collected for each ulna and averaged to obtain a single value for each parameter in each animal. Fluorescent labels were visualized with a Nikon Ti2 microscope at 20× using a Nikon Ri2 monochrome camera, and cortical bone parameters were quantified using BioQuant 2019 Software. Parameters included Ps.MAR and Ps.BFR, as defined by the ASBMR nomenclature guidelines (*Dempster et al., 2013*).

## Western blotting

Western blotting of whole-cell extracts following appropriate treatments or extracts isolated from murine long bone was done as previously described (*Moorer et al., 2017*). Briefly, equal amounts of protein were loaded on SDS–PAGE gels, electrophoresed, and transferred to PVDF membranes. Membranes were blocked in 5% nonfat dry milk and 3% BSA in phosphate-buffered saline with 0.1% Tween-20 for all sclerostin blots. All other blots were blocked in 5% nonfat dry milk in phosphate-buffered saline with 0.1% Tween-20. Primary antibodies dilutions were as follows: sclerostin (1:500); GAPDH (1:2500); β-actin (1:5000), α-tubulin (1:2000); pro-collagen type 1 (Col1a1) (1:1000); pCaMKII (1:1000), total CaMKII (1:1000), p62/sequestosome-1 (1:500), and LC3B (1:500). Antibodies were detected using horseradish peroxidase-conjugated secondary antibody (1:1000–5000) (Cell Signaling Technology) and visualized with enhanced chemiluminescence reagent (Bio-Rad) and analyzed with ImageLab software (Bio-Rad) or fluorescent Licor secondary antibodies (1:20,000) were used and visualized with Licor Oddysey CLx and analyzed using Image Studio v5.

## Statistical analysis

Experiments were repeated a minimum of three times with triplicate samples unless indicated otherwise. For dynamic histomorphometry experiments, animals were randomized into groups by weight. Mice were assigned a random number, and the experimenters were blinded to animal treatment and experimental group during collection and analysis period. Graphs show means, with error bars indicating SD or SEM, as indicated. All statistical test applied to experimental data were carried out in GraphPad Prism 8.0. Data were compared with two-tailed unpaired t-tests, with a two-way ANOVA with Holm–Sidak post hoc correction, or Kruskal–Wallis with Dunn's post hoc test, as indicated. A p-value of <0.05 was used as a threshold for statistical significance.

## Acknowledgements

The Ocy454 cells were provided by P Divieti-Pajevic (Boston University) through support from the Center for Skeletal Research Core at Massachusetts General (NIH P30 AR075042).

## Additional information

### Competing interests

James S Lyons, Joseph P Stains: Holds two patents related to this work. One for the custom fluid shear device used for these experiments (US Patent No US 2017/0276666 A1) and a second for the targeting microtubules (part of this mechano-transduction pathway) to improve bone mass (US Patent No US 2019/0351055 A1). Ramzi J Khairallah: Has a patent pending on colchicine analogs to treat musculoskeletal disorders (PCT/US2018/038300). Ramzi J. Khairallah is affiliated with Myologica, LLC. The author has no financial interests to declare. Christopher W Ward: Holds two patents related to this work. One for the custom fluid shear device used for these experiments (US Patent No US 2017/0276666 A1) and a second for the targeting microtubules (part of this mechano-transduction pathway) to improve bone mass (US Patent No US 2019/0351055 A1). Another patent pending on colchicine analogs to treat musculoskeletal disorders (PCT/US2018/038300). The other authors declare that no competing interests exist.

## Funding

| Funder | Grant reference number | Author |
|---|---|---|
| National Institutes of Health | AR071614 | Christopher W Ward<br>Joseph P Stains |
| Maryland Stem Cell Research Fund | 2018-MSCRFD-4246 | Ricardo A Feldman |
| American Heart Association | 19POST34450156 | Humberto C Joca |
| National Institutes of Health | AR071618 HL142290 | Christopher W Ward |
| National Institutes of Health | GM008181 | Nicole R Gould<br>James S Lyons |
| National Institutes of Health | AR007592 | Katrina M Williams |

The funders had no role in study design, data collection and interpretation, or the decision to submit the work for publication.

## Author contributions

Nicole R Gould, Conceptualization, Data curation, Formal analysis, Validation, Investigation, Visualization, Writing - original draft, Writing - review and editing; Katrina M Williams, Conceptualization, Data curation, Investigation, Writing - original draft; Humberto C Joca, Investigation, Methodology; Olivia M Torre, Conceptualization, Formal analysis, Validation, Investigation, Writing - review and editing; James S Lyons, Conceptualization, Investigation, Writing - review and editing; Jenna M Leser, Validation, Investigation, Writing - review and editing; Manasa P Srikanth, Resources, Investigation; Marcus Hughes, Investigation; Ramzi J Khairallah, Conceptualization, Resources, Supervision, Methodology, Writing - original draft, Writing - review and editing; Ricardo A Feldman, Conceptualization, Resources, Writing - review and editing; Christopher W Ward, Conceptualization, Supervision, Funding acquisition, Investigation, Writing - original draft, Writing - review and editing; Joseph P Stains, Conceptualization, Supervision, Funding acquisition, Writing - original draft, Project administration, Writing - review and editing

## Author ORCIDs

Katrina M Williams http://orcid.org/0000-0003-3729-0630
Olivia M Torre http://orcid.org/0000-0003-3405-6259
Joseph P Stains https://orcid.org/0000-0002-1610-4694

## Ethics

Animal experimentation: All of the animals were handled according to protocol approved by the Animal care and Use Committee at the University of Maryland School of Medicine (Protocol Numbers, 0617013 and 0520007).

## Decision letter and Author response

Decision letter https://doi.org/10.7554/eLife.64393.sa1
Author response https://doi.org/10.7554/eLife.64393.sa2

# Additional files

## Supplementary files

• Source data 1. This is the raw data for the entire manuscript. Each tab of the excel references a different figure.

• Transparent reporting form

## Data availability

All data generated or analyzed during this study are included in the manuscript and supporting files.

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
