## [Decision Letter]

**Acceptance summary:**

Despite a well-established role for sclerostin in mechanical loading-induced bone formation, surprisingly little is known about how biomechanical load to bone leads to decreased sclerostin protein abundance in the osteocytes. In this elegant study, the authors have systematically examined the concept that two distinct, clinically relevant bone anabolic cues (mechanical load and parathyroid hormone) regulate sclerostin degradation post-translationally by directing sclerostin protein to the lysosome using in vitro, ex vivo and in vivo mode The study is elegantly designed, clearly communicated, and rigorously conducted. Additional data and the authors revisions are found to be appropriate in addressing the recommendations of the reviewers. The identified concept mechanical loading and PTH regulate osteocyte sclerostin protein levels acutely by the lysosome can be exploited to develop new drug targets to improve bone mass in conditions such as osteoporosis

**Decision letter after peer review:**

Thank you for submitting your article "Disparate Bone Anabolic Cues Activate Bone Formation by Regulating the Rapid Lysosomal Degradation of Sclerostin Protein" for consideration by *eLife*. Your article has been reviewed by 3 peer reviewers, and the evaluation has been overseen by a Reviewing Editor and Mone Zaidi as the Senior Editor. The following individuals involved in review of your submission have agreed to reveal their identity: Gabriel L Galea (Reviewer #1); Tamara N Alliston (Reviewer #2).

The reviewers have discussed the reviews with one another and the Reviewing Editor has drafted this decision to help you prepare a revised submission.

Summary:

The article by Gould et al. breaks new ground by demonstrating a role for lysosomal-mediated degradation in the mechanosensitive repression of Sclerostin levels in bone. Though the post-translational repression of Sclerostin has long been apparent, no one has yet unraveled the mechanisms. Therefore, this discovery is important to the skeletal biology community – both because of the findings themselves, and because the conditions/models used by this team to make these discoveries will be useful for other investigators, including their ability to manipulate and observe the rapid lysosome-dependent control of Sclerostin levels in vitro and in vivo in response to PTH or mechanical stimulation. In addition to the importance within this field, the work has broad impact on multiple levels including (a) the clinical relevance for understanding and potentially treating osteoporosis and the skeletal phenotypes in individuals with lysosomal disease, and (b) the mechanoregulation of lysosomal function and its relationships to crinophagy, which has implications not only for the regulation of Sclerostin, but also for other factors in and beyond the skeleton (RANKL, insulin).

Essential revisions:

The study is elegantly designed, clearly communicated, and rigorously conducted. However, the reviewers require additional data to support the overall conclusion on the significance of lysosome-mediated degradation of sclerostin in skeletal biology.

First, it is important to elaborate on what gives the authors confidence that the inhibitors were effective and act as expected throughout the study – but especially Bafilomycin A1 and Apocynin in vivo. If BafA1 and Apocynin treatment in vivo work as expected, they should prevent the rapid load-dependent repression of Sclerostin levels (shown in Figure 1D).

Second, the authors demonstration of mechanical load-dependent changes in sclerostin localization in osteocytes lysosomes in vivo by immunohistochemistry would be important to support the in vivo relevance of this pathway in the acute regulation of sclerostin levels. While the western blotting of mechanically loaded mouse ulnas showing previously-undocumented acute reductions in lysate sclerostin levels is interesting, it is unclear if these changes are caused by mechanical loading-induced lysosomal function.

---

## [Author Response]

Essential revisions:The study is elegantly designed, clearly communicated, and rigorously conducted. However, the reviewers require additional data to support the overall conclusion on the significance of lysosome-mediated degradation of sclerostin in skeletal biology.First, it is important to elaborate on what gives the authors confidence that the inhibitors were effective and act as expected throughout the study – but especially Bafilomycin A1 and Apocynin in vivo. If BafA1 and Apocynin treatment in vivo work as expected, they should prevent the rapid load-dependent repression of Sclerostin levels (shown in Figure 1D).

As requested, we performed new experiments using ulnar load in vivo (Figure 6E and Figure 7D, respectively). In these experiments, we showed that the acute effects of a single dose of drug (apocynin to inhibit NOX2 or Bafilomycin A1 to inhibit lysosomes) prior to a single bout of ulnar loading prevented downregulation of sclerostin protein in bone. These data fully support our proposed mechanisms of action and complement the previous data that chronic dosing of these drugs impaired bone formation, as well.

Additionally, new data in Figures 2B and 6A support that activation of this pathway results in the rapid loss of sclerostin protein ex vivo, confirming our mechanisms of action. Together, these data confirm that sclerostin is rapidly degraded in situ by load and that inhibiting early or late stages of the pathway, not only decreases load induced bone formation but also prevents the load induced loss of sclerostin protein in bone.

Second, the authors demonstration of mechanical load-dependent changes in sclerostin localization in osteocytes lysosomes in vivo by immunohistochemistry would be important to support the in vivo relevance of this pathway in the acute regulation of sclerostin levels. While the western blotting of mechanically loaded mouse ulnas showing previously-undocumented acute reductions in lysate sclerostin levels is interesting, it is unclear if these changes are caused by mechanical loading-induced lysosomal function.

As requested, to link the loss of sclerostin protein following acute ulnar loading to the osteocyte, we performed immunofluorescence microscopy on loaded bones that were fixed, sectioned, stained, and counted percent positive osteocytes in these sections (Figure 1 – supplement 1A). As expected, we observed that a single bout of loading results in the rapid loss of sclerostin protein in osteocytes (% sclerostin positive osteocytes). These data mirror those attained with western blotting of bone extracts. I still contend western blotting of this bone derived protein is a far more sensitive and quantitative assessment for tracking sclerostin than immunofluorescence.

We could not co-localize sclerostin and lysosomes in situ as suggested due to technical obstacles. Namely, our lysosome staining tools (e.g., siR-Lysosome or Lysotracker) failed to work in fixed bones processed for sectioning and antibody staining. We tried several methods of detecting lysosomes in fixed, permeabilized bone sections but were unable to detect specific signal. Regardless, we think the new experiments in Figures 6E, Figure 7D, and Figure 1- supplement 1A present a convincing body of evidence that sclerostin downregulation occurs in osteocytes in situ (new data) and that inhibitors targeting distinct aspects of a mechano-signaling pathway (NOX2 or lysosomes, respectively) both block the downregulation of sclerostin in bone (new data) in addition to blunting load induced bone formation (in previous version).